

# A bilateral employment situation prediction model for college students using GCN and LSTM

Junxia Shen

Employment Guidance Division, Luohe Medical College, Luohe, Henan, China

## ABSTRACT

Due to the prevailing trend of globalization, the competition for social employment has escalated significantly. Moreover, the job market has become exceedingly competitive for students, warranting immediate attention. In light of this, a novel prognostic model employing big data technology is proposed to facilitate a bilateral employment scenario for graduates, aiding college students in promptly gauging the prevailing social employment landscape and providing precise employment guidance. Initially, the focus lies in meticulously analyzing pivotal aspects of college students' employment by constructing a specialized employment platform. Subsequently, a classification model grounded in a graph convolution network (GCN) is built, leveraging big data technology to comprehensively comprehend graduates' strengths and weaknesses in the employment milieu. Furthermore, based on the outcomes derived from the comprehensive classification of college students' qualities, a college students' employment trend prediction method employing long and short term memory (LSTM) is proposed. This method supplements the analysis of graduates' employability and enables accurate forecasting of college students' employment trends. Empirical evidence substantiates that my proposed methodology effectively evaluates graduates' comprehensive qualities and successfully predicts their employment prospects. The achieved F-values, 82.45% and 69.89%, respectively, demonstrate the efficacy of anticipating the new paradigm in graduates' dual-line employment.

## INTRODUCTION

Motivated by society's progressive nature, graduates' employment landscape undergoes annual variations (*Cellini & Turner, 2019*; *Fu & Wang, 2021*). The competition for graduate employment has intensified, resulting in heightened societal pressures (*Cellini & Turner, 2019*; *Fu & Wang, 2021*). In recent years, the emergence of new industries, such as artificial intelligence and automation technology, has considerably impacted the employment prospects of college students. While certain occupations with low automation levels have been phased out, these developments have ushered in novel employment opportunities within other domains (*Changxin, 2021*). Moreover, propelled by the ongoing advancements in globalization, international employment prospects for college students have expanded significantly. The demand for multinational enterprises and industries

Corresponding author
Junxia Shen,
sjx_1111112023@163.com

engaged in multilateral trade has surged (*Changxin, 2021*). In light of these factors, this article aims to aid college students in comprehending the ever-evolving landscape of bilateral employment and facilitates forecasting employment trends. To accomplish this, the research delves into the dynamics of bilateral jobs for graduates and presents an employment situation prediction model leveraging big data technology.

With the support of external employment situations and big data, the prediction of college students' employment situation can be realized. It can help college students better plan their careers. Through the analysis and research of future employment trends and market demand, college students can better understand the employment prospects of their majors and which skills and knowledge will be valued more in the future (*Ye, 2020*). College students' employment prediction can also help them make more informed career decisions. Understanding the future employment situation and trends in a highly competitive job market can help college students make more reasonable decisions when choosing majors, internships and employment opportunities to improve their competitiveness and employment opportunities (*Zhu et al., 2022*). In addition, college students' employment prediction can also help the government and educational institutions better plan educational resources and policies. Understanding the future employment situation and trends can help the government and educational institutions better formulate education policies to cultivate talents more in line with market demand (*Cheng, 2022*). Adopting big data to master the changes in the employment situation has become an efficient method for society, colleges and students to obtain real-time information (*Gong, 2019*). Mastering the huge social employment situation data and the past year's employment information for regular data storage, dispersed in different nodes, improve the reliability and availability of data. Then, data mining, deep learning and other technologies are used to professionalize these meaningful data. The data can be split into small pieces and allocated to different nodes for parallel processing to improve processing efficiency and realize real-time processing and analysis. Relying on the advantages of distributed computing, the computing tasks are distributed to multiple computing nodes for parallel computing to improve the computing speed and efficiency.

To help college students master the employment situation and make reasonable judgments about employment goals according to their conditions when looking for jobs, I carry out research on the new situation prediction model of bilateral employment of graduates by big data technology to realize the improvement and innovation of college students' employment environment. First, I propose a college student employment information and employment prediction system by constructing a data layer, base layer, decision layer and other structures in order to realize the real-time evaluation of college students' employment. Then, relying on the above system, I put forward a classification model of college students' comprehensive quality to accurately guide college students' employment goals. Finally, I combined the complete quality classification model and proposed a college student employment situation prediction model, which conducts the calculation of regional and enterprise information, to achieve an accurate forecast of college student employment situation. The main contributions of this article are:

1. A classification model of graduates' comprehensive quality by big data is proposed to evaluate the comprehensive quality and suitability for college students' jobs.

2. A prediction method of graduates' job trends based on deep learning is proposed to assist college students in evaluating their comprehensive quality and recommend employment directions according to the evaluation.

## LITERATURE REVIEW

### Research status of graduates' employment

With the development of the economy, the employment problem of college students is becoming more and more serious. The research shows that the employment of college students mainly has the following characteristics. Firstly, most students find it challenging to find and determine their career goals (*Simón, Díaz & Costa, 2017*) actively. Secondly, the gap between college students' actual and expected income after graduation has widened, and the return on education investment is far lower than college students' expectations (*Chu, Liu & Liu, 2019*). Finally, the employment of college students has a solid regional (*Zuo & Gao, 2022*). Therefore, studying the employment of college students has become a hot topic. Research on the graduates' employment is usually based on demand, supply and supply-demand matching. In addition, personal career planning and social practice experience also have a specific impact on college students' employment. Different ideas have different effects. Early establishment of individual career planning can better find a suitable position, and social practice experience is the basis for the successful employment of college students. Students, community service, all kinds of part-time jobs, and other social practice activities to actively participate in and improve their work experience to make the future work like a fish in water. *Boudarbat (2008)* believes that voluntary unemployment is the main reason for the unemployment of college students. *Cholwe (2008)* discussed the problem of university student unemployment in Zambia and thought that the leading causes of unemployment were lack of demand and structural contradictions. *Chuang (1999)* established a model and believed that college students' characteristics and career search variables were the decisive factors affecting college students' employment. *Choi, Jeong & Hwa Jung (2005)* analyzed the problem of unemployment caused by the low return rate of college students' education in South Korea. *Mok & Montgomery (2021)* believes that the lack of training for college students and the mismatch between supply and demand are the main reasons for the unemployment of Spanish college students. *Pongton & Suntrayuth (2019)* believes that the employment status of graduates is related to the rise and fall of the job market.

Therefore, employment guidance has been carried out in various countries and regions, and almost all colleges and universities have employment guidance agencies with complete facilities and systems (*Stavrou, 2022*). As a result, a relatively mature theory has been formed in the career development and guidance theory. It also includes career decision-making and employment guidance theory (*Hossen, Chan & Hasan, 2020*). For example, Parsons' trait factor theory (*Jones, 1994*), Holland's Personality type Theory (*Niles, 1993*) and Shupa's theory (*Watanabe-Muraoka, Takeshi Senzaki & Herr, 2011*). *Folsom &*

*Reardon (2003)* introduced and expounded the trait factor theory by analyzing Parsons's outstanding contribution to career planning and guidance. *Niles (1993)* confirmed that similar Holland personality types were beneficial to career guidance. Japanese scholars further expounded on the influence of career development theory on young students (*Reichling & Wulf, 2009*). *Haekal & Muttaqien (2021)* comprehensively and deeply analyzed the characteristics of career guidance courses in the United States, which inspired the construction of career guidance courses in China. Career guidance continues to develop in the West and gradually presents a trend of standardization and curriculum.

## Research status of big data mining technology

Big data brings the rapid development of data mining. Whether it is industry, academia, business, or the service industry, historical information is stored. Technological advances adopt big data. In this process, people often have different understandings and views on data mining, but generally speaking, it can be described from two aspects: technical level and commercial application. On the technical level, data mining uses software to extract the required data from the database of the studied party, mine and analyze it, and then obtain some hidden, unknown, and valuable knowledge in the data. This knowledge can be displayed in various forms, such as decision trees, rule sets, graphs, formulas, and models. In terms of commercial application, data mining uses algorithms to model and analyze the extracted data, make decisions, and predict the results to support enterprises' development and realize commercial value.

Scholars have conducted data mining work on employment analysis in many aspects, including predicting career changes. For example, people's knowledge is predicted using person tags and data records inside enterprises (*Herrmann et al., 2023*). In studying people's career changes, a series of determining factors are found behind the career changes, and the regular and reproducible patterns are mined (*Sisavath, 2023*). In addition, literature (*Ashaye et al., 2023*) explicitly studies people's historical work records and uses machine learning algorithms to recommend new jobs for people. Iowa State University (*Bastedo, Altbach & Gumport, 2023*) in the United States has a career management service system in which the career guidance center objectively evaluates students' career-matching status through the measurement of questionnaires and scales. Students can also get online guidance from professionals to rationally and efficiently plan and decide the suitable career development direction. Research and prediction on the graduate job market have become an important part of career guidance in Western universities (*Millar et al., 2023*). Career guidance activities can reflect the information feedback process at a macro level, so the information can help colleges and universities adjust talent training plans and then improve the effectiveness of the overall career guidance.

## AN EMPLOYMENT SITUATION FORECASTING MODEL BY BIG DATA TECHNOLOGY

The employment of college graduates is a system engineering, which is related to a variety of factors, such as the reputation of the university itself, the major of the students, the current economy, the learning situation of the students, the expectations of the students for

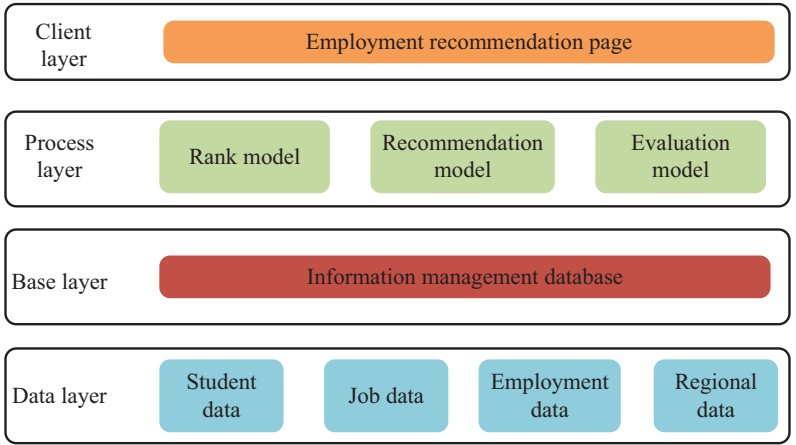

**Figure 1 Environment creation program for infant.** The system architecture of our entire information management and employment recommendation system in shown, which is composed of data layer, base layer, decision-making layer and front-end web page.

the graduation unit, *etc.*, with solid time-varying and diversity, which brings specific difficulties to the job forecasting of graduates. The essence of the problem of college graduate employment prediction is to fit the change between the factors and graduates' employment through a certain method, find the characteristics of college graduate employment change, and predict the trend of college graduate employment change according to the factors. In addition, using big data to analyze the number of jobs. We have used different factors, *e.g.*, selected cities or regions, including industry information, regional job saturation, *etc.*, to provide detailed data support for college students and the most affordable and intimate service.

Therefore, I propose a college students employment information management and employment prediction system. The system architecture of my entire information management and employment recommendation system is shown in Fig. 1, which comprises a data layer, base layer, decision-making layer and front-end web page. Among them, the data layer is used to collect student data (including personal information, course selection records, test scores, job resumes, *etc.*), job data (recruitment company information, recruitment career information, recruitment requirements, *etc.*), and employment data (employment data of students in my school over the years), which will be important data sources for later applications. After the data modeling of the base layer, the processing layer uses a variety of models for data cleaning, feature extraction, model calculation and evaluation. At the same time, I designed and implemented the evaluation module composed of online and offline assessments. The offline evaluation calculates the recommendation results' precision, recall and other evaluation indicators. Considering that the employment information of college students is closely related to their own quality, regions and enterprises. Meanwhile, the information is sequential. I use the comprehensive quality of college students, regional information and enterprise information in every year to reflect the differences in college students' employment.

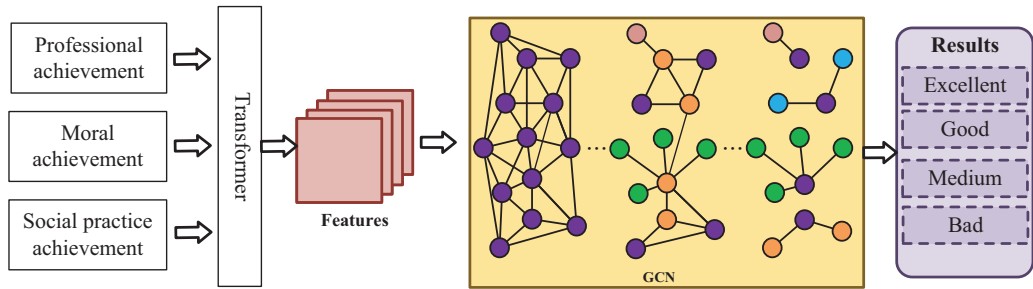

**Figure 2 Classification model of graduates' comprehensive quality.** Through the comprehensive evaluation of students' professional performance, moral education performance, social practice performance and other data, we realize the judgment of college students' comprehensive quality, and provide data sources for the prediction of college students' employment situation, as shown in the figure.

## Classification model of college students' comprehensive quality

The comprehensive quality of college students is the comprehensive ability of knowledge, skills, attitudes and values. In the university stage, in addition to mastering professional knowledge and skills, students should also have efficient learning ability, be able to learn independently and constantly explore, and continuously improve their learning efficiency and learning ability. At the same time, college students should have innovative thinking and creative ability, be able to think independently and solve problems, put forward new ideas and programs, and have a sense of innovation. They should have good teamwork skills when completing tasks and be able to communicate and cooperate effectively with others to achieve common goals. College students should also be able to actively participate in social welfare activities, care about social issues, and contribute to social development. Understand the industry development trend and career requirements, master vocational skills and practical experience, adapt to the workplace environment and show their professional ability. In addition, college students should have good psychological qualities, be able to cope with challenges and pressure and have the ability to self-regulate and receive psychological counseling.

Therefore, through the comprehensive evaluation of students' professional performance, moral education performance, social practice performance and other data, I realize the judgment of college students' comprehensive quality and provide data sources for the prediction of college students' employment situation, as shown in Fig. 2.

Firstly, I extract the sequence features of college students' quality and form a similar feature matrix by the obtained feature relationship. Then, the feature matrix and transformer are used to compute the relationship between various quality features of graduates. Finally, the quality features were used as the expansion information of graph nodes and the information of graph nodes to form a graph convolutional neural network (GCN) (Zhao et al., 2019), as shown in the yellow box in Fig. 2. Because there are connections between various quality features of students, these connections can be expressed in a topology structure in the current scenario. Therefore, I use GCN to construct a topology and fill the parent node and multiple child nodes of GCN with different quality information and sub-modules. For example, if the score of specialized

courses is the parent node, then the score of different kinds of courses is the child node. The output results of the graph convolutional network were added to the fully connected layer to obtain the classification of college students' comprehensive quality. The core calculation formula of the transformer is as follows:

$$\hat{z}_i = MSA(\ln(z_{i-1})) + z_{i-1} \tag{1}$$
$$z_i = MLP(\ln(\hat{z}_i)) + \hat{z}_i \tag{2}$$

where LN refers to the layer normalization, MSA means the multi-head self-attention and MLP denotes the multilayer perceptron.

Each multi-head attention module of the transformer consists of a self-attention mechanism. Self-attention is an adaptive attention mechanism. Constructing a multi-head attention module with several self-attentions can dig deep into the correlation between features and highlight key information in the data. By embedding self-attention into the model, I can significantly extract the key points of college students' comprehensive quality and promote the model to identify the grade of college students' comprehensive quality. where $W^*$ is the weight matrix and Z is the weighted feature vector, the calculation process is as follows:

$$Q = FW^Q \tag{3}$$
$$K = FW^K \tag{4}$$
$$V = FW^V \tag{5}$$
$$Z = Softmax\left(\frac{QK^T}{\sqrt{d_k}}\right)V \tag{6}$$

## Employment situation prediction model based on the comprehensive quality of graduates

To accurately predict the new situation of bilateral employment for college students, after obtaining the classification results of college students' comprehensive quality, we will comprehensively consider the effects of comprehensive quality, the situation of employment target regions and the development prospects of enterprises. Considering the role of the above three kinds of information in the bilateral employment of graduates, the employment situation prediction model based on the comprehensive quality of graduates is proposed.

The three factors of the comprehensive quality of graduates, the situation of the employment target area and the prospect of enterprise development will have different performances with the time change. Therefore, I regard these three factors as a time series, which can be observed using multiple sensors in an equal time interval according to a given sampling rate. I define multiple variable time series data as $T = \{(T_i, c_i), i = 1, 2, \ldots n\}$. The multivariate time series classification task is to predict the class label ci for a given multivariate time series sample Ti with unknown labels. Considering the long-term dependence of LSTM, I calculate the relationship between two different adjacent sequences. Due to the timeliness of the above three factors, I quantified the relationship

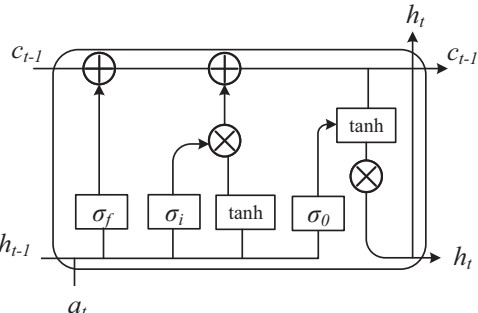

**Figure 3 LSTM structure diagram.** The three convolved time series features are input into the time series prediction model LSTM at the same time, whose structure is shown in the figure.

between different periods of features as the distance between them. Given two subsequences of length l belonging to variables b1 and b2, the Euclidean distance between them is defined as follows.

$$Dist(s, b) = \sqrt{\frac{1}{l}\sum_{k=1}^{l} ||t_s - t_b||^2} \tag{7}$$

For a multivariate time series sample Ti of length L, subsequences s and b, I use a sliding window of length l to extract all subsequences of length l on Ti. Finally, the three convolved time series features are input into the time series prediction model LSTM (*Yu et al., 2019*) simultaneously, whose structure is shown in Fig. 3, and the final prediction results are obtained.

The classification model of college students' comprehensive quality is used. I can quantify college students' professional scores, moral education scores, social practice scores and other data, which can give them a clear self-cognition, to help them have a career orientation in the job hunting process. In addition, using the employment situation prediction model based on the comprehensive quality of college students, through the input region and the specific state of enterprises, can assist college students in carrying out long-term work planning to achieve the employment situation prediction of college students.

# EXPERIMENT AND ANALYSIS

## Dataset and implement details

I used the CORGIS dataset (https://corgis-edu.github.io/corgis/csv/graduates/) to test the effectiveness of my method. The experiments are carried out on a device with i7-13650 Cpu and Rtx 2080 Gpu, the operating system is Linux, and the network model is implemented under the Tensorflow framework. The experimental setup used in my experiment is in Table 1.

**Table 1 Experimental parameter setting.**

| Parameters | |
|---|---|
| Epoch | 50 |
| Batch size | 8 |
| Initial learning rate | 0.0004 |
| Optimizer | Adam |
| Weight attenuation | 0.00001 |
| Momentum | 0.9 |

To evaluate model performance, I use precision, recall, and F-measure as evaluation criteria, which are calculated as follows:

$$Vaule_{Precision} = \frac{TP}{TP + FP} \tag{8}$$

$$Vaule_{Recall} = \frac{TP}{TP + FN} \tag{9}$$

$$Vaule_F = \frac{2 Vaule_{Precision} \times Vaule_{Recall}}{Vaule_{Precision} + Vaule_{Recall}} \tag{10}$$

where TP means the number of correct detected samples, the FP denotes the number of wrong samples and FN refers to the number of detected wrong samples.

## Results and discussion

I conduct experiments on the new situation prediction model of bilateral employment of college students on the CORGIS dataset.

For college students' comprehensive quality classification model, I select some excellent classification models: ResNet (*Wu, Shen & Van Den Hengel, 2019*), GRU (*Dey & Salem, 2017*), transformer (*Han et al., 2021*), Decision Tree (*Song & Ying, 2015*), XGBoost (*Chen et al., 2015*) and Deit (*Touvron et al., 2021*). The comparison results are shown in Fig. 4. My method has obtained the best performance scores in Precision, Recall and F-value of 80.45%, 82.36% and 82.45%, respectively. Compared with ResNet and GRU, the F-value of my method is improved by 6.00% and 10.76%, respectively. Compared with the traditional methods XGBoost and Decision Tree, my method has achieved a comprehensive lead in three evaluation indicators: Precision, Recall and F value. Compared with Transformer-based methods, the F-value of my method exceeds their performance by 2.20% and 0.12%, respectively. In addition, I can conclude that my method has achieved an overall lead in two indicators of Recall and Precision. I adopt the GCN structure to simulate the connection between each key point when considering the comprehensive quality of college students. At the same time, I show the model convergence of my method and other methods at training time, as shown in Fig. 5. My proposed child behavior classification model can complete the training goal of the model faster and reach the state of model parameter convergence in the training process.

In addition, to verify my employment situation prediction model based on the quality of graduates, I conducted ablation experiments on the model. According to the results of
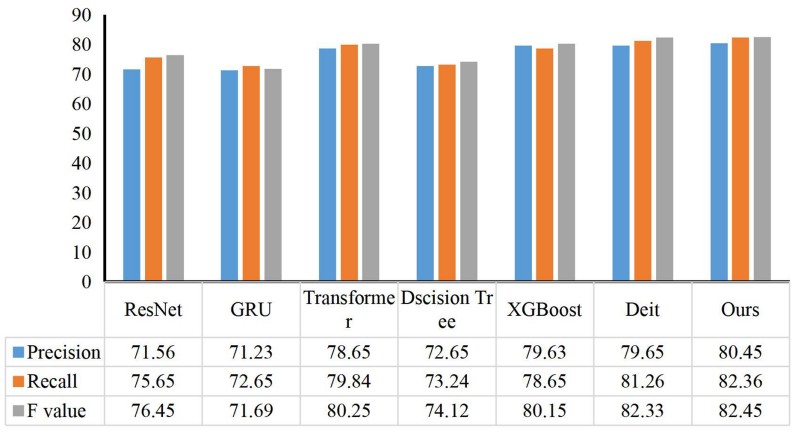

**Figure 4 Comparison between my method and other methods.** For the comprehensive quality classification model of college students, I select some excellent classification models, they are ResNet, GRU, Transformer, Decision tree, XGBoost and Deit.     

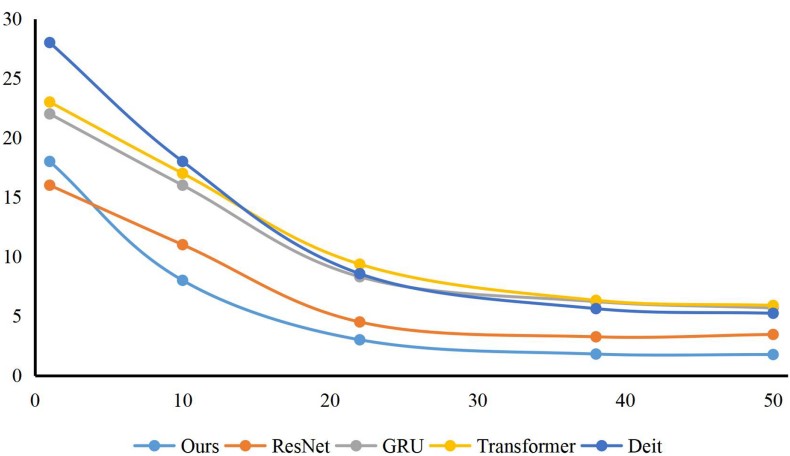

**Figure 5 The training of my method and other methods.** I show the model convergence of our method and other methods at training time is shown. It can be found that our proposed child behavior classification model can complete the training goal of the model faster and reach the state of model parameter convergence in the training process.     

college students' comprehensive quality, the situation of employment target regions and the development prospects of enterprises, different special fusions were carried out respectively. Because I use LSTM as the network infrastructure, the ablation experiment contains a total of the contents, as shown in Table 2. From the table, it can be found that adding different features based on the Baseline can enhance the prediction of the employment situation. In addition, the combination of other forms features can effectively improve the F-score. Finally, the employment situation can be effectively predicted by using the three features of the comprehensive quality results of graduates, the employment target area situation and the development prospect of enterprises.

Table 2 **Ablation experiments.** The table shows that adding different features on the basis of Baseline can enhance the prediction of employment situation.

| Baseline | Comprehensive quality | Employment target areas | Enterprise development prospect | F value |
|---|---|---|---|---|
| LSTM | | | | 62.89 |
| | ▲ | | | 65.23 |
| | | ▲ | | 66.77 |
| | | | ▲ | 65.45 |
| | ▲ | ▲ | | 67.46 |
| | ▲ | | ▲ | 68.26 |
| | | ▲ | ▲ | 67.56 |
| | ▲ | ▲ | ▲ | 69.89 |

Table 3 **Comparison with Deit, XGBoost and Transformer.**

| Methods | Recall | Precision | F value |
|---|---|---|---|
| Deit | 68.67 | 69.08 | 68.76 |
| XGBoost | 68.23 | 70.87 | 68.96 |
| Transformer | 68.89 | 70.93 | 69.32 |
| Mine | 69.23 | 71.35 | 69.89 |

In order to verify my employment situation prediction model based on the comprehensive quality of college students, I compare my method with Deit, XGBoost and Transformer, and the results are shown in Table 3. I can conclude that my method achieves the best results with an F value of 69.89%. In addition, I ranked first in the recall, precision and F value. Thanks to the time series structure, I achieved better recall and precision. This also shows that my method can help college students to predict their employment situation.

## CONCLUSION

I propose a novel bilateral employment situation prediction model based on big data technology to guide college students in their employment journey. This model leverages the employment platform for college students and incorporates a classification model that assesses the comprehensive quality of college students using big data technology. By analyzing regional and enterprise information in conjunction with the comprehensive quality assessment of college students, this article introduces a prediction method based on deep learning to forecast the employment trend of college students. The proposed model enables the accurate prediction of the bilateral employment situation for college students. Experimental results demonstrate that the comprehensive quality classification model successfully facilitates self-assessment for college students. Moreover, the bilateral employment situation prediction model provides precise employment recommendations and prospects for college students. This aids them in making informed decisions and

selecting suitable job opportunities. Overall, the proposed model offers valuable assistance to college students in their job selection and employment endeavors.

### Funding

This work was supported by the Luohe Social Science Planning Leading Group for the 2022 social science research project. The funders had no role in study design, data collection and analysis, decision to publish, or preparation of the manuscript.

### Grant Disclosures

The following grant information was disclosed by the authors:
Luohe Social Science Planning Leading Group.

### Competing Interests

The authors declare that they have no competing interests.

### Author Contributions

- Junxia Shen conceived and designed the experiments, performed the experiments, analyzed the data, performed the computation work, prepared figures and/or tables, authored or reviewed drafts of the article, and approved the final draft.

### Data Availability

The data and code are available in the Supplemental File.

### Supplemental Information

Supplemental information for this article can be found online at http://dx.doi.org/10.7717/peerj-cs.1494#supplemental-information.

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
