# Peer review of "A bilateral employment situation prediction model for college students using GCN and LSTM"

_PeerJ Computer Science, doi:10.7717/peerj-cs.1494_

## Round 0.1 · original submission · Major Revisions

Dear authors,

Your paper has been reviewed by the experts in the field and myself. you will see that they have several concerns to be addressed. Please take care of their comments along with the following suggestions

1. In a separate section, Please outline the study's novelty and its applicability in the real world.
2. There are a couple of structural issues in the grammar of the paper, therefore, please carefully revise it during the revision
3. Please provide a comparison table showing how your work is better than the existing studies (if any)

thank you again

Reviewer 1 ·

Basic reporting

• Authors need to add keywords according to the abstract content;
• The parameters of formula (1) and (2) have not been explained, and the author needs to supplement them;
• Also, other studies should be cited to increase the theoretical background of each of the methods used;
• Findings should be contextualized in the literature and should be explicit about the added value of the study towards the literature;
• The logic of the introduction is a little confusing, and it is suggested that the overall revision be more smooth and organized;
• Page 8 mentions that GCN consists of characteristic information. I don't quite understand what the author means;
• The conclusion is similar to the abstract, and does not highlight the advantages of the model and its future development;
• References from recent years should be added, preferably from good journals.

Experimental design

• The corresponding definition and interpretation can be simply added to the evaluation index P,R and F-measure in the experimental part;

Validity of the findings

• Ablation experiments of the predicted model are not enough to prove the advantages of LSTM, so other comparative experiments should be added to highlight the LSTM model;

Additional comments

Due to the trend of globalization, the competition for social employment is
becoming more and more fierce. Therefore, this paper proposes a new situation
prediction model for bilateral employment of graduates through big data technology, in
order to help college students timely grasp the social employment situation and make
accurate employment guidance. Although the article has made some achievements, it
still has the following deficiencies:
• Authors need to add keywords according to the abstract content;
• The parameters of formula (1) and (2) have not been explained, and the author needs to supplement them;
• The corresponding definition and interpretation can be simply added to the evaluation index P,R and F-measure in the experimental part;
• Also, other studies should be cited to increase the theoretical background of each of the methods used;
• Findings should be contextualized in the literature and should be explicit about the added value of the study towards the literature;
• The logic of the introduction is a little confusing, and it is suggested that the overall revision be more smooth and organized;
• Ablation experiments of the predicted model are not enough to prove the advantages of LSTM, so other comparative experiments should be added to highlight the LSTM model;
• Page 8 mentions that GCN consists of characteristic information. I don't quite understand what the author means;
• The conclusion is similar to the abstract, and does not highlight the advantages of the model and its future development;
• References from recent years should be added, preferably from good journals.

·

Basic reporting

In view of the increasingly tense student employment trend, this paper puts forward a new situation forecast model, through various experiments and analysis, finally puts forward a college student employment trend forecast method, played a role in assisting graduates. But there are some suggestions for the method model presented in this paper:
1. The GCN and LSTM algorithms in the title are suggested to be highlighted in the abstract;
2. The introduction part is too repetitive, should be reorganized to highlight the background introduction and author contribution;

Experimental design

3. I would advise the author to read through the formula and explain clearly the principle and definition of the formula;
4. Literature review lacks support, so more background and papers related to college students' difficulties in graduation can be added;
5. What method does the author use to find out the changing characteristics of college graduates? The text does not seem to elaborate;

Validity of the findings

6. Each multi-head attention module in Transformer includes a self-attention mechanism. What are the reasons and benefits of this?
7. Formula (7) is the Euclidean distance, the author calculated its action is not elaborated.

---

## Round 0.2 · accepted · Accept

Based on the opinion of experts, your paper is recommended for publication. Thank you for your nice contribution.

Reviewer 1 ·

Basic reporting

All corrections have been made and are eligible for publication as such.

Experimental design

All corrections have been made and are eligible for publication as such.

Validity of the findings

All corrections have been made and are eligible for publication as such.

·

Basic reporting

no comment

Experimental design

no comment

Validity of the findings

no comment